

# The antiferromagnetic $S = 1/2$ Heisenberg model on the $C_{60}$ fullerene geometry

**Roman Rausch[1,2⋆], Cassian Plorin[3] and Matthias Peschke[4]**

**1** Technische Universität Braunschweig, Institut für Mathematische Physik,
Mendelssohnstraße 3, 38106 Braunschweig, Germany
**2** Department of Physics, Kyoto University, Kyoto 606-8502, Japan
**3** Department of Physics, University of Hamburg,
Jungiusstraße 9, D-20355 Hamburg, Germany
**4** Institute for Theoretical Physics Amsterdam and Delta Institute for Theoretical Physics,
University of Amsterdam, Science Park 904, 1098 XH Amsterdam, The Netherlands

⋆ r.rausch@tu-braunschweig.de

## Abstract

We solve the quantum-mechanical antiferromagnetic Heisenberg model with spins positioned on vertices of the truncated icosahedron using the density-matrix renormalization group (DMRG). This describes magnetic properties of the undoped $C_{60}$ fullerene at half filling in the limit of strong on-site interaction $U$. We calculate the ground state and correlation functions for all possible distances, the lowest singlet and triplet excited states, as well as thermodynamic properties, namely the specific heat and spin susceptibility. We find that unlike smaller $C_{20}$ or $C_{32}$ that are solvable by exact diagonalization, the lowest excited state is a triplet rather than a singlet, indicating a reduced frustration due to the presence of many hexagon faces and the separation of the pentagonal faces, similar to what is found for the truncated tetrahedron. This implies that frustration may be tuneable within the fullerenes by changing their size. The spin-spin correlations are much stronger along the hexagon bonds and exponentially decrease with distance, so that the molecule is large enough not to be correlated across its whole extent. The specific heat shows a high-temperature peak and a low-temperature shoulder reminiscent of the kagomé lattice, while the spin susceptibility shows a single broad peak and is very close to the one of $C_{20}$.



# 1   Introduction

The C$_{60}$ buckminsterfullerene molecule, where 60 carbon atoms sit on the vertices of a truncated icosahedron, is a prominent molecule with a wealth of chemical and nanotechnological applications [1–3], and is also of interest in terms of correlated-electron physics. A lattice of C$_{60}$ molecules becomes superconducting when doped with alkali metals [4–7], with a critical temperature of around 40K. This is unusually high for a typical phononic mechanism, so that an electronic mechanism that results from an on-site Hubbard interaction $U$ is under discussion as well [8, 9]. At half filling (no doping), a strong $U$ is well-known to cause electron localization via the Mott mechanism and the resulting low-energy properties are described by the antiferromagnetic spin-1/2 Heisenberg model

$$H = J \sum_{\langle ij \rangle} \mathbf{S}_i \cdot \mathbf{S}_j, \tag{1}$$

where $\mathbf{S}_i$ is the spin operator at site $i$, $J = 4t^2/U > 0$ is the exchange integral and $t$ is the hopping integral between nearest-neighbour sites $i$ and $j$.

However, the prototypical Mott systems are transition metal oxides with strong Coulomb repulsion in a narrow $d$-band, while in carbon atoms, we are dealing with a valence $p$-band. As a consequence, while the nearest-neighbour hopping parameters are estimated around 2−3 eV, the Hubbard repulsion $U$ is estimated to be around 9 eV [10–12], which would place the system into the intermediate-coupling range. Still, since solving the full Hubbard model for 60 orbitals on a 2D-like geometry is a hard problem, we may attempt to understand the Heisenberg approximation first. Other authors have argued that there should only be a quantitative difference [12], since the system is finite. The Hartree-Fock solution shows a phase transition to magnetic order at $U_c/t \approx 2.6$ [13]. This seems to indicate that local moments may be already well-formed for a fairly small $U$. As soon as they are formed, mean field is biased towards an ordered solution, but we expect the exact ground state of this finite system to always be a singlet.

Apart from trying to approximate the Hubbard model, a spin model on a fullerene-type geometry is interesting on its own, being connected to the problem of frustrated spin systems. These arise on non-bipartite geometries like the triangular, kagomé or pyrochlore lattice, with building blocks of three-site clusters that cannot accommodate antiferromagnetic bonds in a commensurate fashion. This tends to induce spin-liquid states that are disordered and non-trivial [14–21]. In fullerenes, we instead find 12 pentagon clusters that are also frustrated due

to the odd amount of sites. This has no strict correspondence in the 2D plane, since a tiling by regular pentagons is not possible. However, a Cairo tiling is possible by irregular pentagons, resulting in two bonds $J$ and $J'$ [22]. While non-bipartite, this lattice can be divided into two inequivalent sublattices, tends to show ferrimagnetic order, and is thus quite different from our case [22].

A frustrated spin system is still quite challenging for a theoretical description. For example, the infamous sign problem [23] inhibits an efficient simulation with the Quantum Monte Carlo technique. However, tensor-network approaches do not suffer from it. $C_{60}$ is in particular well-suited to a solution using the density-matrix renormalization group (DMRG) [24] due to its finite and very manageable amount of sites.

The truncated icosahedron is part of the icosahedral group $I_h$. To its members belong two of the Platonic solids, the icosahedron with 12 sites and the dodecahedron with 20 sites (which is also the smallest fullerene $C_{20}$) [25]. The former has only triangular plaquettes, the latter only pentagonal ones, and both are small enough to be solved by full diagonalization if spatial symmetries are exploited to reduce the Hilbert space size [26]. $I_h$ also has 5 members within the Archimedean solids, of which the icosidodecahedron with 30 sites (triangular and pentagonal faces) has been the subject of particularly intense study [27–31], since this is the geometry of the magnetic atoms in the Keplerate molecules {$Mo_{72}V_{30}$}, {$Mo_{72}Cr_{30}$} and {$Mo_{72}Fe_{30}$}, with $S = 1/2, 3/2$ and $5/2$ respectively [32–34]. It is solvable by exact diagonalization for $S = 1/2$ [27]. Small fullerenes up to $C_{32}$ can also be solved by exact diagonalization [35,36], but have different symmetries. Finally, the truncated tetrahedron is a 12-vertex Archimedean solid, which is not a member of $I_h$, but has a geometry that is similar to $C_{60}$ [13,37,38], consisting out of four triangles separated by hexagons. For this reason, it is often also counted as a fullerene $C_{12}$. All these smaller molecules offer a very useful comparison and benchmark.

Each fullerene $C_n$ contains $n/2 - 10$ hexagons and 12 pentagons [39], so that for $n \geq 44$ the number of hexagon faces starts to dominate. For $n \to \infty$, we can expect that the fullerene properties approach those of a hexagonal lattice. But without undertaking the full calculation, it is impossible to say where exactly the crossover happens or what properties might be retained in the large-$n$ limit. In fact, the small fullerenes up to $C_{32}$ do not behave monotonously [35]: For example, the ground-state energy for $C_{26}$ and $C_{28}$ is larger than for $C_{20}$ and the first excited state for $C_{28}$ is a triplet instead of a singlet.

In this paper, we present the solution of the Heisenberg model on the $C_{60}$ geometry. Previous works treated the problem classically [12] or approximately [23], while our calculation is very precise for the ground state. Jiang and Kivelson solved the $t-J$ model on $C_{60}$ [8], which should coincide with our result at half filling. However, they discussed very different questions; and we further present results for the lowest excited states as well as thermodynamics.

Due to two dissimilar types of nearest-neighbour bonds, the corresponding hopping integrals may be slightly different, $t_1 \approx 1.2\ t_2$, leading to different exchange couplings $J_1 \neq J_2$ [12,40]. For simplicity, we ignore this fact and use a homogeneous $J = J_1 = J_2$ for all bonds. The correlations along the bonds turn out to be nonetheless very different as a consequence of the geometry, as will be seen below. We take $J = 1$ as the energy scale, giving all energies in units of $J$ and all temperatures in units of $J/k_B$, where $k_B$ is the Boltzmann constant.

## 2 Ground state and correlation functions

### 2.1 Technical notes

Our code incorporates the spin-SU(2) symmetry of the model following Ref. [41], which reduces both the bond dimension of the matrix-product state (MPS) representation of the wave-

function and the matrix-product operator (MPO) representation of the Hamiltonian. The latter can be further reduced using the lossless compression algorithm of Ref. [42]. It gives only a small benefit of 8% reduction for $H$ itself, with the resulting maximal MPO bond dimension of $\chi(H) = 35 \times 32$ (from $38 \times 35$). The benefit for $H^2$ is larger, yielding $\chi(H^2) = 564 \times 468$ (reduced from $1444 \times 1225$, hence by 55%). With these optimizations, the ground state can be found quite efficiently and we can take the variance per site

$$\Delta E^2 / L = \left(\langle H^2 \rangle - E^2\right)/L, \tag{2}$$

as a global error measure that is immune to local minima.

Since DMRG requires a linear chain of sites, we must map the $C_{60}$ vertices onto it, which creates long-range spin-spin interactions across it. The important factors to consider are: 1. the maximal hopping range (the bandwidth of the corresponding graph), 2. the average hopping range, 3. the fact that DMRG is particularly good for nearest-neighbour bonds on the chain, so that a representation where the sites $i$ and $i+1$ are connected should be beneficial (this will also be practical for finite-temperature calculations further below). Our mapping is an infalling spiral on the Schlegel diagram, such that the first and last site have maximal distance, and is shown in Fig. 4. We have also tried out the mapping of Jiang and Kivelson [8] and a graph compression using the Cuthill-McKee algorithm [43]; and find similar MPO compression and ground state convergence results. A random permutation of the sites, on the other hand, leads to a representation with a large MPO bond dimension which the compression algorithm is unable to decrease, and the convergence becomes much worse. For a benchmark with a system solvable by exact diagonalization, we compare a similar spiral mapping for the icosidodecahedron with the mapping used by Exler and Schnack [29,44] and find that both approaches come within 99.97% of the exact $S = 1/2$ ground-state energy [22] at a bond dimension of $\chi_{SU(2)} = 500$. Thus we conclude that as long as the numbering of the sites is reasonable and more or less minimizes the hopping distances, the dependence on the numbering itself is small and an inaccuracy that results from a suboptimal numbering can simply be compensated by moderately increasing the bond dimension. This is in line with the conclusions of Ummethum, Schnack and Läuchli [29]. Finally, we note that by checking the energy variance (Eq. 2) and the distribution of spin-spin correlations at a given distance (see Sec. 2.3), we have good independent error measures.

Interestingly, we find that the number of required subspaces per site in the DMRG simulation is similar to the Heisenberg chain (around seven), but each subspace requires large matrices (with $3500 \sim 4000$ rows/columns, see Tab. 1). This makes the simulation very memory-intensive, requiring several hundred GB of RAM for good precision.

## 2.2 Energy

The ground state lies in the singlet sector with $S_{\text{tot}} = \sum_i \langle \mathbf{S}_i \rangle = 0$ (see Tab. 1). The energy per spin is found to be $E_0/L = -0.51886$. This is lower than the previous result of $E_0/L = -0.50798$ obtained by a spin-wave calculation on top of the classical ground state [23].

Looking at the change in ground-state energy with molecule size, we may compare with the truncated tetrahedron $C_{12}$ ($E_0/L = -0.475076$), $C_{20}$ ($E_0/L = -0.486109$) and $C_{32}$ ($E_0/L = -0.4980$ [35]), and recognize that the value indeed slowly approaches the one for the hexagonal lattice $E_0/L \approx -0.55$ [45]. On the other hand, it is quite close to the much smaller icosahedron ($E_0/L = -0.515657$) which has the same icosahedral symmetry, but only contains triangular plaquettes. Finally, the icosidodecahedron has the highest energy $E_0/L = -0.441141$ [22], probably due to the strong frustration.

Table 1: Properties of the ground state and the lowest eigenstates: total energy $E$, energy density $E/L$, the gap to the ground state, the total spin $S_{\text{tot}}$, the full bond dimension $\chi_{\text{SU(2)}}$ with spin-SU(2) symmetry, the maximal bond dimension of the largest subspace $\chi_{\text{sub,SU(2)}}$, the effective bond dimension $\chi_{\text{eff}}$ that would be required when not exploiting the symmetry, the energy variance per site (Eq. 2), and the overlap with the ground state.

| $E$ | $E/L$ | gap | $S_{\text{tot}}$ | $\chi_{\text{SU(2)}}$ | $\chi_{\text{sub,SU(2)}}$ | $\chi_{\text{eff}}$ | $\Delta E^2/L$ | GS overlap |
|---|---|---|---|---|---|---|---|---|
| -31.131(7) | -0.51886(1) | - | 0 | 10000 | 3966 | 43146 | $8 \cdot 10^{-5}$ | - |
| -30.775(6) | -0.51292(7) | 0.356(0) | 1 | 10000 | 3770 | 44302 | $1.9 \cdot 10^{-4}$ | 0 |
| -30.440(9) | -0.50734(9) | 0.690(8) | 0 | 10000 | 3582 | 46846 | $1.6 \cdot 10^{-4}$ | $\sim 10^{-8}$ |
| -30.3(2) | -0.505(3) | 0.8(2) | 2 | 5000 | 1855 | 24546 | $1.4 \cdot 10^{-3}$ | 0 |

## 2.3 Correlation functions

The truncated icosahedron is an Archimedean solid, so that all of its sites (vertices) are equivalent; but since two hexagons and one pentagon come together at a vertex, there are two different nearest-neighbour bonds: one that is shared between the two hexagons and two that run between a pentagon and a hexagon (with the total count of 30 and 60, respectively, see Fig. 3 and Fig. 4). We shall call them "hexagon bonds" (H-bonds) and "pentagon bonds" (P-bonds). The wavefunction must respect this geometry, but as the mapping to a chain introduces a bias, this only happens for a sufficiently large bond dimension. Thus, we can average over the respective bonds and take the resulting distribution width as a measure of error, with a $\delta$-distribution expected in the limit of $\chi \to \infty$. Figure 1 shows the result for distances up to $d = 4$, from which we see that for the given bond dimension, the distributions have already become sufficiently sharp.

Similarly, we have up to five distinct types of bonds for the remaining distances $d = 2-9$. In the numerics, they can be distinguished as distinct peaks in the distribution of the correlations $\langle \mathbf{S} \cdot \mathbf{S}_d \rangle$ (Figs. 1 and 2). Up to $d = 4$ we classify them by a sequence of H- and P-bonds. For example, at $d = 2$ we have two PP-bonds by going along the P-bonds twice, ending up in the same-face pentagon of a given vertex; and four HP-bonds, by going along H and P (in any order), ending up in the same-face hexagon (see Fig. 3).

A striking pattern is that the path that can be labelled by alternating H- and P-bonds has the strongest correlations at each $d$. Such a path is possible up to $d = 7$; and up to $d = 3$, it ends in the same-face hexagon. Hence, it seems that since the hexagons are not frustrated, putting a lot of correlation into these bonds can lower the energy more effectively. In fact, the sequence of intrahexagon values is closely matched by the infinite Heisenberg chain [46] or the $L = 6$ Heisenberg ring. On the other hand, the bonds involving pentagons are closely matched by the values of the dodecahedron. Figure 5 shows a comparison. As a consequence of this, the ground-state energy can actually be naively approximated by taking $E_0 \approx 30 \langle \mathbf{S} \cdot \mathbf{S}_{d=1} \rangle [\text{chain}] + 60 \langle \mathbf{S} \cdot \mathbf{S}_{d=1} \rangle [\text{dodecahedron}] \approx -32.739$, coming within 95% of the precise DMRG value.

Finally, we also note that for $d = 5, 6, 7$ the correlations acquire mixed signs and for $d = 8, 9$ the staggered antiferromagnetic order is flipped, i.e. we have $\langle \mathbf{S} \cdot \mathbf{S}_{d=8} \rangle < 0$ and $\langle \mathbf{S} \cdot \mathbf{S}_{d=9} \rangle > 0$.

Overall, the pattern is very similar to the truncated tetrahedron, where the maximal distance is $d = 3$, the stronger correlations are also found for the same-face hexagon bonds; and a mixed sign is acquired for $d = 3$ (see Fig. 5).

Looking at the decay of the correlations with distance, we find $\xi \sim 1.4$ when an exponential fit $\left| \langle \mathbf{S} \cdot \mathbf{S}_d \rangle \right| \sim \exp(-d/\xi)$ is applied to the maximal absolute values and $\xi \sim 1.2$ if it is applied to bond-averaged values (see inset of Fig. 5) Previously, $\xi = 3 \sim 4$ was proposed [23] based

Table 2: Comparison of the singlet and triplet gaps for various polyhedra with $L$ vertices. The smaller value is underlined.

| polyhedron | $L$ | singlet gap | triplet gap |
|---|---|---|---|
| trunc. tetrahedron ($C_{12}$) | 12 | 0.896 | 0.688 |
| icosahedron | 12 | 0.533 | 0.900 |
| dodecahedron ($C_{20}$) | 20 | 0.316 | 0.514 |
| icosidodecahedron | 30 | 0.047 | 0.218 |
| trunc. icosahedron ($C_{60}$) | 60 | 0.691 | 0.356 |

on a strong-coupling Quantum Monte Carlo study of the single-band Hubbard model. The truncated tetrahedron and the dodecahedron have larger excitation gaps, but the maximum possible distance is $d = 3$ and $d = 5$, respectively, so that they are correlated over practically their whole extent (see Fig. 5). The icosidodecahedron has a small gap, but the behaviour of the correlations is very similar to the dodecahedron. An exponential fit does not give good results for these small molecules. For $C_{60}$, the smallest gap is actually about as large as for the dodecahedron, but the maximal distance is $d = 9$ and the drop-off across the whole molecule is larger. In this sense, the $C_{60}$ spin state is disordered.

The fullerenes $C_n$ have a kind of thermodynamic limit $n \rightarrow \infty$, where we expect that the magnetic properties should approach the properties of the hexagonal lattice with Néel order [47], which should be detectable by large spin-spin correlations in a finite system. Clearly, we are still far away from that limit: The pentagons disrupt the bipartiteness and lead to a disordered state instead.

# 3 Lowest triplet and singlet excitations

By fixing $S_{\text{tot}} = 1$, we can compute the lowest excited state in the triplet sector and look at its properties as well. We limit ourselves to the expectation value of the local spin $\langle \mathbf{S}_i \rangle$ and the nearest-neighbour correlation functions. The values of $\langle \mathbf{S}_i \rangle$ are shown in Fig. 6. We observe that a good part of the angular momentum (about 60%) localizes on a 20-site ring along a "meridian" of the molecule. As this breaks the spatial symmetry, we conclude that the lowest $S_{\text{tot}} = 1$ state is degenerate beyond the three components of the spin projection, i.e. has a multiplicity $> 1$ of its irreducible point group representation. The symmetry should be restored when averaging over the whole degenerate subspace.

The icosahedral group has the irreducible representations $A$ (1), $T$ (3), $F$ (4) and $H$ (5) [48], where the brackets indicate the multiplicity. The members of the icosahedral group that are solvable by exact diagonalization behave as follows: The lowest triplets of the icosahedron transform as $T_{2g}$, $T_{1u}$ and $T_{2u}$; of the dodecahedron as $T_{2g}$, $F_u$, $T_{2u}$ [26]; and of the icosidodecahedron as $A_g$, $H_u$, $H_g$ [22]. Hence, while the lowest triplet is 3-fold degenerate for the former two, it is nondegenerate for the latter. Our results indicate that the lowest triplet of $C_{60}$ is again degenerate.

Finding out its irreducible representation requires either to work within a symmetry-adapted basis or to construct the whole multiplet of excited states. Since the degeneracy is at least 3-fold, we would need at least the lowest four eigenstates in the $S = 1$ sector to a precision that is smaller than the gap to the next triplet state. Since the degeneracy of the excited states is of no crucial physical importance, we do not attempt this procedure in our work.

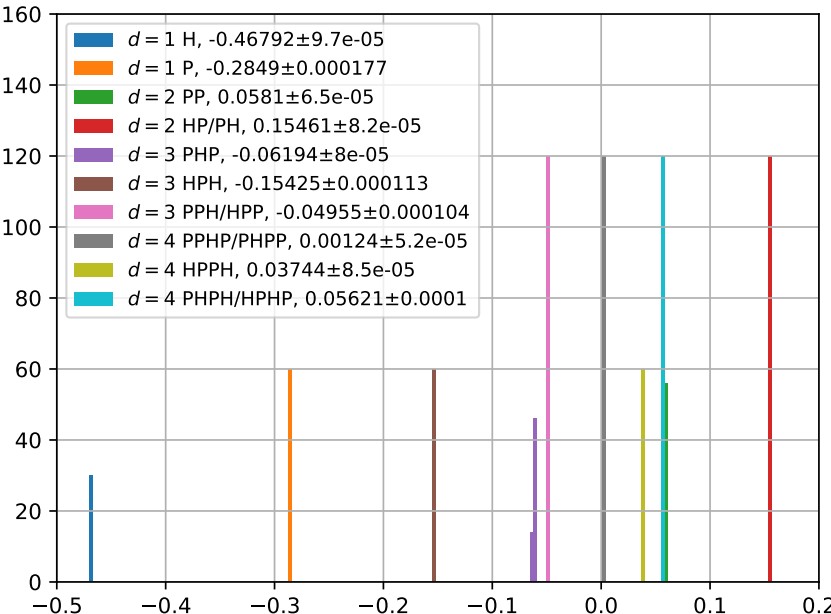

Figure 1: Histogram of the spin-spin correlation function $\langle \mathbf{S} \cdot \mathbf{S}_d \rangle$ in the ground state for distances $d = 1$ to $4$ and the various types of $C_{60}$ bonds. For the meaning of the labels, see Fig. 3 and the explanation in the text. The standard deviation of the distribution is taken as the error measure in the legend. The binsize is 0.003.

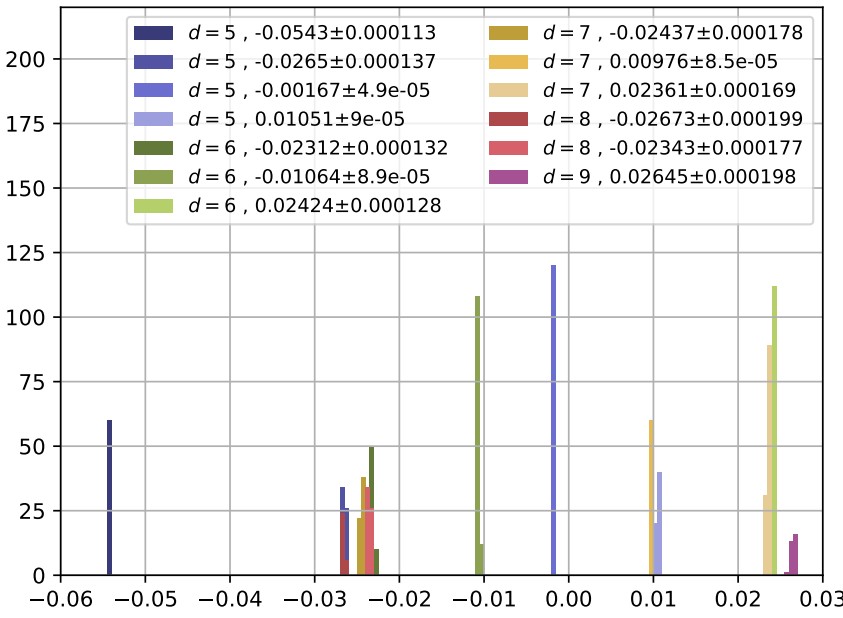

Figure 2: Histogram of the spin-spin correlation function $\langle \mathbf{S} \cdot \mathbf{S}_d \rangle$ in the ground state for distances $d = 5$ to $9$ and the various types of $C_{60}$ bonds. The standard deviation of the distribution is taken as the error measure in the legend. The binsize is 0.0005.

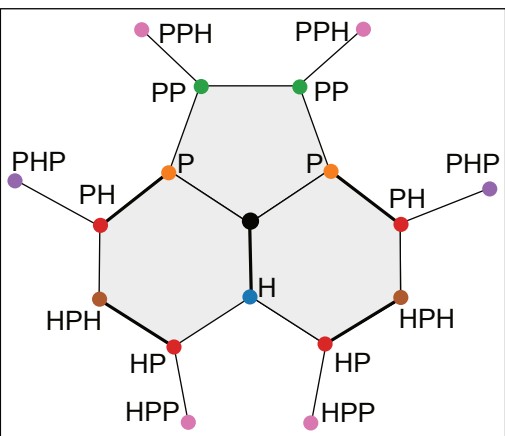

Figure 3: Neighbourhood of a given site (black circle) showing the various types of bonds (cf. Fig. 1).

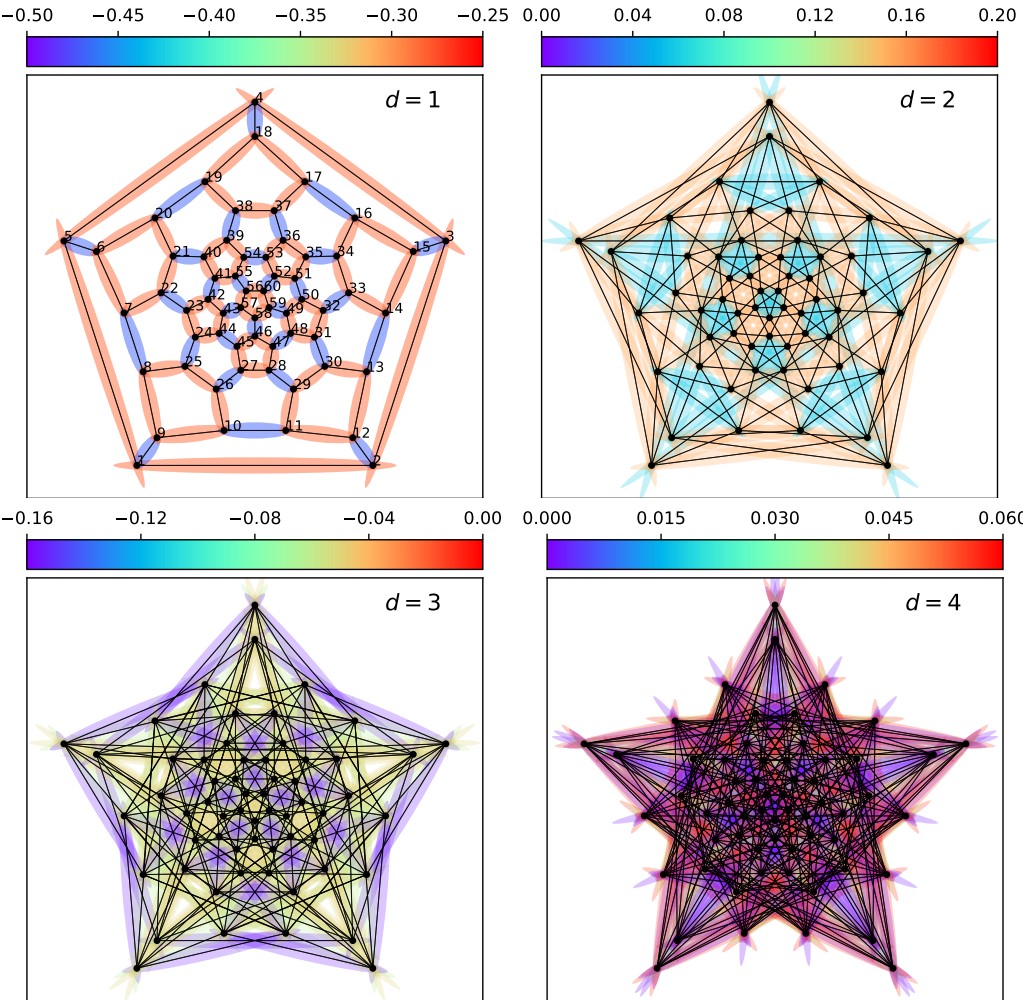

Figure 4: Visualization of the spin-spin correlation function $\langle \mathbf{S} \cdot \mathbf{S}_d \rangle$ in the ground state for distances $d = 1, 2, 3, 4$ in real space on the planar Schlegel projection of $C_{60}$. The plot for $d = 1$ also shows the chosen enumeration of the sites.

We find that the symmetry-breaking 20-site ring is remarkably robust in our DMRG simulation and arises from different random starting states and for different site enumerations. In a

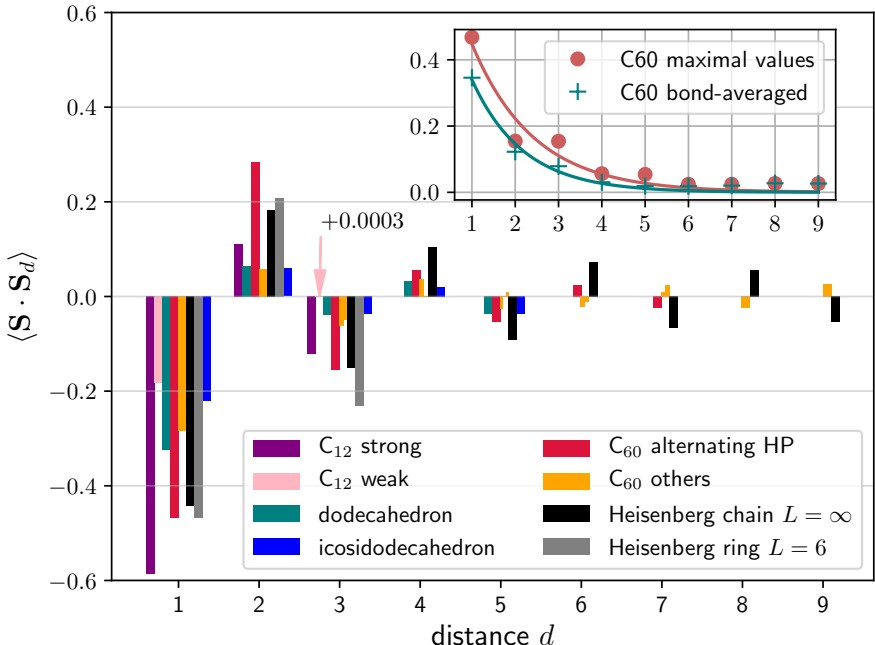

Figure 5: Comparison of the spin-spin correlation function between different geometries: analytical values for the infinite Heisenberg chain [46], numerically exact values for the $L = 6$ Heisenberg ring, $C_{12}$ (truncated tetrahedron) and the dodecahedron [26]. The icosidodecahedron values are according to our own DMRG calculation. The $C_{60}$ alternating HP bonds are formed by alternating jumps along H and P (cf. Fig. 3), starting with H; and link two sites within a hexagon up to $d = 3$. Note that the icosidodecahedron has two inequivalent bonds for $d = 2, 3, 4$, but the correlation along the second-type bond is very small and is omitted. The weak bond for $C_{12}$ at $d = 3$ is +0.0003 and thus barely visible. The inset shows an exponential fit for the distance dependence of the $C_{60}$ spin-spin correlations, either by taking the maximal values for each $d$ or by taking bond-averaged values.

realistic setting, we expect that the spatial symmetry would in any case be at least slightly broken by the Jahn-Teller effect, so that such a state may split from the degenerate subspace. In fact, for doped $C_{60}$, one observes the same preference for a localization of the excess electron along a 20-site ring [49], whereas in our case this happens to a doped spin (excess angular momentum).

A striking property of Heisenberg spins on smaller icosahedral molecules [22, 26], as well as for smaller fullerene geometries [35], is that the first excited state is not a triplet, but rather a singlet, a signature of frustration connected to spin-liquid behaviour [14, 50–52]. The icosidodecahedron has in fact a large amount of singlet states below the first triplet [22]. On the other hand, for the truncated tetrahedron, the first excited state is a triplet for $S = 1/2$.

We therefore calculate the first excited state in the singlet sector ($S_{\text{tot}} = 0$) as the lowest state of the Hamiltonian $\tilde{H} = H + E_p |E_0\rangle\langle E_0|$ with a sufficiently large energy penalty $E_p > 0$ that must be larger than the neutral gap. The result is shown in Tab. 1. The neutral gap $\Delta_{S=0} = E_1(S_{\text{tot}} = 0) - E_0(S_{\text{tot}=0}) = 0.691$ turns out to be significantly larger than the singlet-triplet gap $\Delta_{S=1} = E_0(S_{\text{tot}} = 1) - E_0(S_{\text{tot}=0}) = 0.356$ (cf. the other polyhedra in Tab. 2). We attribute this behaviour to the reduced frustration of the $C_{60}$ molecule due to the large amount of hexagonal faces. Furthermore, we note that all the pentagonal faces are completely separated by the hexagons, so that regions with adjacent frustrated pentagons that are present in smaller fullerenes are broken up in $C_{60}$.

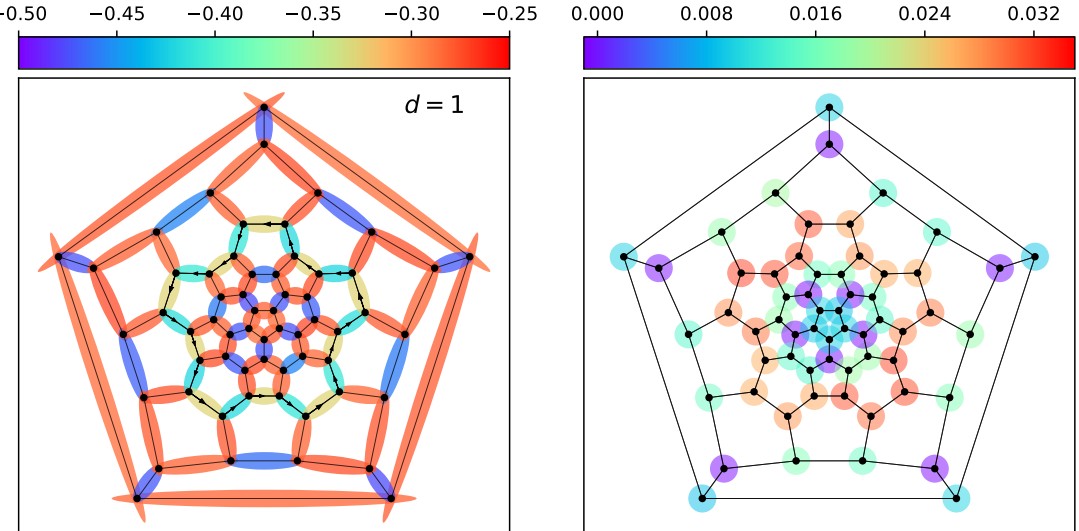

Figure 6: Left: Visualization of the nearest-neighbour spin-spin correlations $\langle \mathbf{S} \cdot \mathbf{S}_{d=1} \rangle$ in the lowest triplet state, $S_{\text{tot}} = 1$. The 20-site ring of altered correlations is highlighted with arrows. Right: Visualization of the local spin $\langle \mathbf{S}_i \rangle$ in the same state.

Looking at the spin-spin correlations in the $S_{\text{tot}} = 0$ excited state in Fig. 7, we note that the singlet excitation is also characterized by a 20-site ring with altered correlations, albeit differently positioned. Once again, this indicates degeneracy and we can compare to the small molecules. Starting from the ground state, the lowest singlets of the icosahedron and dodecahedron both transform as $A_u, H_g, A_g$ [26]; and of the icosidodecahedron as $A_g, A_u, T_{1u}$ [22]. The former two show 5-fold degenerate excited singlets, while the latter shows a nondegenerate one; and $C_{60}$ is once again more similar to the smaller molecules.

## 4  Thermodynamics

### 4.1  Technical notes

We incorporate finite temperatures into the DMRG code using standard techniques [53]. By doubling the degrees of freedom, we go from a description using the wavefunction to a description using the density operator. This density operator is again purified into a state vector, but all operators act on the physical sites only, so that the additional "ancilla" sites are automatically traced over when taking expectation values using the state $\left|\beta\right\rangle = \exp\left(-\beta H/2\right)\left|\beta = 0\right\rangle$. The entanglement entropy between the physical sites and the ancillas becomes equal to the thermal entropy. Finally, we can initiate the state at infinite temperature $\beta = 1/T = 0$ by taking the ground state of the entangler Hamiltonian

$$H_{\beta=0} = \sum_i \mathbf{S}_i \cdot \mathbf{S}_{a(i)}, \tag{3}$$

where $a(i)$ indicates the ancilla site attached to the physical site $i$.

We then apply a propagation in $\beta$ using the TDVP (time-dependent variational principle) algorithm [54] with a step size of $d\beta = 0.1$. At each time step we have the choice of whether to

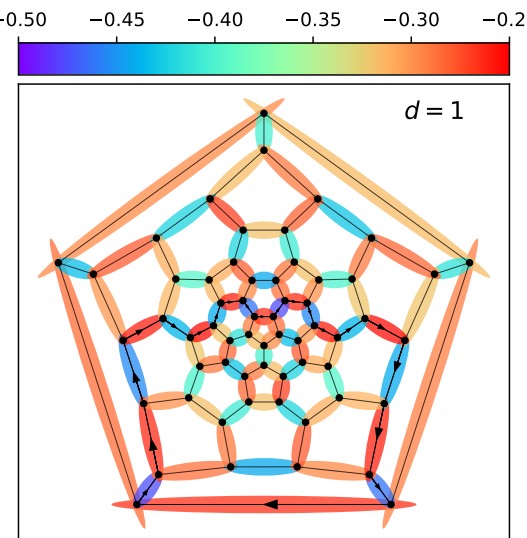

Figure 7: Visualization of the nearest-neighbour spin-spin correlations $\langle \mathbf{S} \cdot \mathbf{S}_{d=1} \rangle$ in the first excited singlet state, $S_{\text{tot}} = 0$. The 20-site ring of altered correlations in the lower part is highlighted with arrows.

apply the two-site algorithm which allows to dynamically grow the bond dimension from the initial product state; or the one-site algorithm which does not increase the bond dimension, but is much faster. Since an upper limit must be in any case set on the bond dimension in the calculations, it is no longer useful to use the two-site algorithm once it saturates. At this point we switch to the faster one-site algorithm (typically around $\beta = 6 - 10$). There is also the technical question of whether to incorporate the ancillas as separate sites (at the cost of longer-ranged hopping) or as "super-sites" [53]. We take the super-site approach for better accuracy.

The TDVP algorithm is known to get stuck in a product state without being able to build up the initial entanglement [21]. We find that this happens whenever the sites $i$ and $i + 1$ are not connected by a nearest-neighbour bond. Since we have chosen super-sites and a numbering where $i$ and $i + 1$ are always connected (see Sec. 2.1), this problem does not appear in our computations.

To strike a balance between accuracy and running time, we can limit the bond dimension *per subspace* to $\chi_{\text{sub,SU(2)}} \sim 300 - 600$, rather than limiting the total bond dimension. This ensures that the largest matrix is at most $\chi_{\text{sub,SU(2)}} \times \chi_{\text{sub,SU(2)}}$ and the duration of the remaining propagation can be estimated. The downside is that the resulting $\chi_{\text{SU(2)}}$ at each site does not in general correspond to the $\chi_{\text{SU(2)}}$ lowest singular values and has to be seen as an order-of-magnitude estimate. A benchmark of this approach for the numerically solvable $C_{20}$ is given in Appendix A. Table 3 shows the parameters that were used in the thermodynamic calculations.

The relevant quantities are the partition function

$$Z_\beta = \langle \beta | \beta \rangle, \tag{4}$$

the internal energy

$$E(\beta) = \langle H \rangle_\beta = Z_\beta^{-1} \langle \beta | H | \beta \rangle, \tag{5}$$

the specific heat per site (or per spin):

$$c(T) = \frac{C(T)}{L} = \frac{1}{L} \frac{\partial E}{\partial T} = \frac{1}{L} \beta^2 \left[ \langle H^2 \rangle_\beta - \langle H \rangle_\beta^2 \right], \tag{6}$$

Table 3: Parameters of the thermodynamic calculations. For an explanation of the bond dimensions, see Tab. 1. The underlined values were fixed.

| $\chi_{\text{sub,SU(2)}}$ | $\chi_{\text{SU(2)}}$ | $\chi_{\text{eff}}$ |
|---|---|---|
| 300 | $\sim 2000$ | $\sim 16000$ |
| 400 | $\sim 3000$ | $\sim 20000$ |
| 600 | $\sim 4700$ | $\sim 36700$ |
| 1163 | 3000 | $\sim 13600$ |
| 1507 | 4000 | $\sim 17500$ |

and the zero-field uniform magnetic susceptibility

$$\chi = \frac{1}{L} \lim_{\mathbf{B}\to 0} \nabla_{\mathbf{B}} \cdot \mathbf{M} = \frac{1}{L} \beta \big[ \left\langle \mathbf{S}^2 \right\rangle_\beta - \left\langle \mathbf{S} \right\rangle_\beta^2 \big], \tag{7}$$

where $\mathbf{M}$ is the magnetization at a given external field strength $\mathbf{B}$ and the Hamiltonian is changed to $H \to H - \mathbf{B} \cdot \mathbf{S}$, with the total spin $\mathbf{S}$:

$$\mathbf{M} = \left\langle \mathbf{S} \right\rangle_{\mathbf{B},\beta} = Z_{\mathbf{B},\beta}^{-1} \left\langle \beta = 0 \middle| \mathbf{S}\, e^{-\beta(H - \mathbf{B} \cdot \mathbf{S})} \middle| \beta = 0 \right\rangle. \tag{8}$$

While the specific heat could be exactly calculated using the squared Hamiltonian average $\left\langle H^2 \right\rangle_\beta$, in practice this becomes quite expensive at every $\beta$-step, so we use a numerical differentiation of $E(\beta)$ with spline interpolation instead.

## 4.2 Specific heat

The result for $c(T)$ is shown in Fig. 8 and is compared to smaller molecules that exhibit a two-peak structure: For the truncated tetrahedron they are so close to each other that they cannot be resolved, while being distinct for the dodecahedron. For $C_{60}$, we find instead a high-$T$ peak (around $T \sim 0.58$) and low-$T$ shoulder (around $T \sim 0.15 - 0.19$). The high-$T$ peak can be attributed to the energy scale given by $J = 1$ and is a general feature of Heisenberg chains [53, 55, 56]. The low-$T$ peak can be attributed to the second scale of the energy gap. This can also be compared to the specific heat of the icosidodecahedron, which has three peaks [28, 30, 31]. The middle peak points to the presence of another gap in the region of low-energy states which is absent in the other systems.

We recall that for a two-level system given by the Hamiltonian $H = \text{diag}(0, \Delta)$, the specific heat has a Schottky peak at $T/\Delta \approx 0.417$. In other words, a maximum appears when the temperature is tuned to the middle of the gap $\Delta$. This is roughly consistent with the gap values given in Tab. 1. The fact that we have a shoulder rather than a clear peak implies that several states of close energy contribute to $c(T)$, i.e. a comparatively high density of states close to the first excited state. In fact, we can see that as the bond dimension in the DMRG calculation is increased, we are able to better describe the low-lying states, leading to a flattening of a very shallow peak to a shoulder. Furthermore, we can say that the states in this vicinity must be singlets or triplets, since the quintet gap lies even higher (see Tab. 1).

The icosidodecahedron has been called "kagomé on a sphere" [27], since both geometries have corner-sharing triangles, and several attempts have been made to relate the two systems to each other [30, 31]. The low-energy properties of the kagomé lattice are not entirely clear, however: Some results point to a gapped state with a singlet-triplet gap of 0.13 and a very small neutral gap of $\sim 0.05$ [18, 19, 57], others to a gapless phase [58–62]. In the case of a molecule, a fair comparison should in any case be to a finite kagomé plaquette that has a finite-size gap.

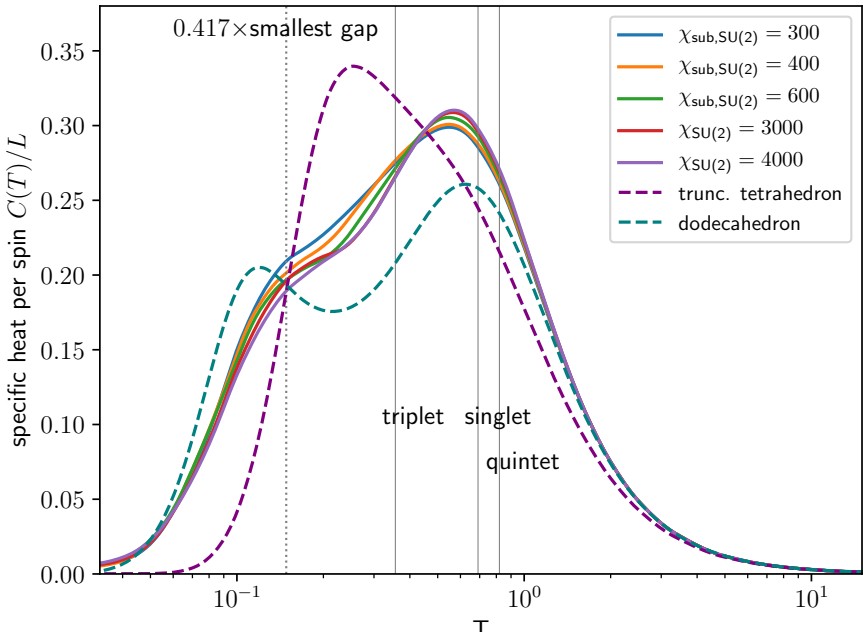

Figure 8: Specific heat of $C_{60}$ for different bond dimensions (Eq. 6). The grey vertical lines indicate the triplet, singlet and quintet gaps with respect to the ground state. The dotted vertical line indicates 0.417 times the lowest gap (triplet). For the parameters, compare Tab. 3.

The low-temperature behaviour of the specific heat is consequently also very difficult to establish. What is well-established is the position of the main peak at $T \approx 0.67$ [16, 63, 64] and a shoulder below it at $T \sim 0.1-0.2$. At a very small $T \sim 0.01$ another peak is found in a finite system which moves up to merge with the shoulder as the system size is increased [64], while tensor-network calculations directly in the thermodynamic limit show no such peak in the first place [63]. The shoulder below the main peak is remarkably similar to the shape that we obtain for $C_{60}$. We also note that the main peak lies below $T = 1$ for both $C_{60}$ and the kagomé lattice, while it is above $T = 1$ for the icosidodecahedron [28]. Since the specific heat is a function of eigenvalues only, we may wonder if the geometry of six triangles around a hexagon of the kagomé lattice leads to a similar eigenvalue distribution as for $C_{60}$ (which has three pentagons around each hexagon) for singlet and triplet excitations that contribute around $T \sim 0.1-0.2$. On the other hand, the large number of singlets close to the kagomé ground state [65] is clearly better matched by the strongly frustrated icosidodecahedron.

## 4.3 Spin susceptibilty

Figure 9 shows the result for the susceptibility $\chi(T)$. It can be interpreted in a similar way, the difference being that singlet states do not contribute anymore. Moreover, it is easy to show that for high temperatures, $\chi(T)$ follows a universal Curie law $\chi(T) \sim 3/4 \cdot T^{-1}$, while for $T \to 0$ we expect $\chi \to 0$, since the ground state is a spin singlet and not susceptible to small fields. In between, $\chi(T)$ should have at least one peak. We observe that it is positioned at a higher temperature for the truncated tetrahedron due to the larger singlet-triplet gap (see Tab. 2). The dodecahedron and $C_{60}$, on the other hand, are remarkably close, though $\chi(T)$ tends to be slightly larger for $C_{60}$ and does not go to zero as fast for very small temperatures, which we ascribe to the smaller singlet-triplet gap.

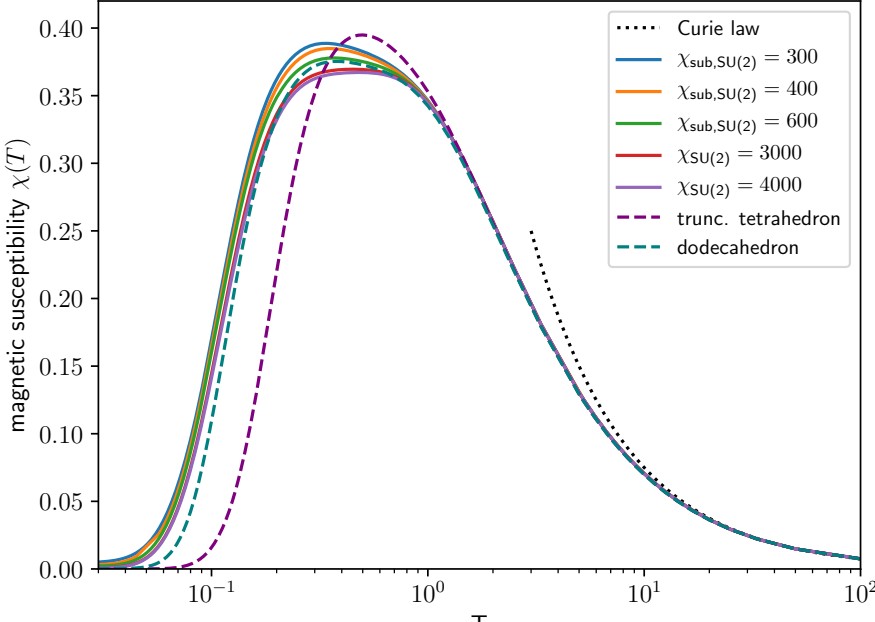

Figure 9: Zero-field uniform magnetic susceptibility $C_{60}$ (Eq. 7) for different bond dimensions. Parameters as in Fig. 8.

## 5 Conclusion

We have presented a solution of the Heisenberg model on the $C_{60}$ fullerene geometry. The spin-spin correlations in the ground state can be determined very accurately using DMRG and indicate that the $C_{60}$ molecule is large enough not to be fully correlated across its full extent. The strongest correlations are found along an alternating path of hexagon and pentagon bonds, a consequence of the fact that the hexagons are not frustrated. Furthermore, for large distances, we find a deviation from the staggered sign pattern of an antiferromagnet.

Most strikingly (and unlike smaller fullerenes), the first excited state is a triplet and not a singlet, indicating weaker frustration. This can be attributed to the large number of unfrustrated hexagon faces, suggesting that frustration is tuneable in small fullerenes as a function of their size. Still, we find that the ground state of $C_{60}$ is disordered with a very short correlation length of $\xi \approx 1.2 \sim 1.4$.

Thus, taking the point of view of the pentagons we can say that the frustration is significantly lowered because all the pentagonal faces are separated from each other by hexagons. On the other hand, taking the point of view of the hexagons we can say that a Néel-like state is prevented by the perturbing pentagonal faces, and one would need larger fullerenes to approach the honeycomb lattice limit.

In terms of thermodynamics, we find a two-peak structure of the specific heat, similar to what is found for the dodecahedron or the kagomé lattice down to $T \sim 0.1 - 0.2$. The low-temperature feature is very shallow for $C_{60}$, forming a shoulder, which indicates relatively densely lying singlet and triplet excited states. The spin susceptibility shows a broad peak very similar to the dodecahedron, but approaches zero less rapidly for $T \to 0$.

All the properties of $C_{60}$ are quite different from the icosidodecahedron, which is highly frustrated, with many low-energy singlet states, a non-degenerate first excited singlet and

triplet, as well as a three-peak structure in the specific heat. On the other hand, we observe much similarity to the truncated tetrahedron: Most notably, the lowest excited state is a triplet and the spin-spin correlations follow the same pattern of being stronger for the same-face hexagon and acquiring a mixed sign for large distances.

We have not attempted to find out the spatial symmetry transformations of the lowest eigenstates, but can conclude that the ground state is non-degenerate, while the first excited singlet and triplet are degenerate, based on the breaking of spatial symmetries or its absence. Another open question is whether the frustrated pentagons can still measurably affect any properties of $C_n$ in the large-$n$ limit. DMRG is well equipped to answer these questions and solve the Heisenberg model for even larger $n$, or for fullerene dimers [66]. Another system that is well-suited for DMRG is the encapsulation of magnetic rare-earth atoms by fullerenes or fullerene-like molecules [67–69], inasfar these can be simulated by the Heisenberg model.

We also attempted to solve the full Hubbard model on the $C_{60}$ geometry, but find that the variance per site is several orders of magnitude higher, so that one would also require an even higher bond dimension at a higher numerical complexity of twice the local Hilbert space size, while probably still tolerating larger errors. An improvement that can bring us closer to the Hubbard case at half filling could in principle be achieved by including higher orders in $1/U$. Up to $\mathcal{O}(U^{-3})$, we have $J = 4t^2/U - 16t^4/U^3$ and a next-nearest-neighbour term $J' = 4t^4/U^3$ [70–72] which may again increase frustration. At $\mathcal{O}(U^{-5})$ a biquadratic term is induced whose inclusion would be quite difficult.

Another intriguing question is how the properties of the undoped $C_{60}$ are related to the results of Jiang and Kivelson [8] that show an attractive pair binding in the doped system. Such pair binding is also present for the truncated tetrahedron [37,73], is weak for the cube [37], but absent for the dodecahedron [73]. This means that one has to study excitations that result from removing electrons within the t-J model, rather than flipping spins. One can hope to relate the attractive pair binding to a geometrical feature like the weak frustration (which $C_{60}$ and the truncated tetrahedron have in common) and perhaps establish a picture that is analogous to the famous relation between resonating valence bond states and superconductivity [74].

# Acknowledgements

**Funding information** R.R. thanks the Japan Society for the Promotion of Science (JSPS) and the Alexander von Humboldt Foundation. Computations were performed at the PHYSnet computation cluster at Hamburg University. R.R. gratefully acknowledges support by JSPS, KAKENHI Grant No. JP18F18750. C.P. is supported by the Deutsche Forschungsgemeinschaft (DFG) through the Cluster of Excellence Advanced Imaging of Matter – EXC 2056 – project ID 390715994. M.P. has received funding from the European Research Council (ERC) under the European Union's Horizon 2020 research and innovation programme (Grant Agreement No. 677061).

# A   Specific heat of $C_{20}$

As a benchmark of the thermal DMRG algorithm, we calculate the specific heat of the dodecahedron and show the results in Fig. 10. While the ground state offers no challenge for DMRG and converges in a matter of seconds, the $\beta$-propagation is more demanding and we see that a high bond dimension is required to get the precise location and height of the low-temperature peak. However, even smaller bond dimensions are able to qualitatively capture the general two-peak structure. The implication for $C_{60}$ is that while we cannot claim that the

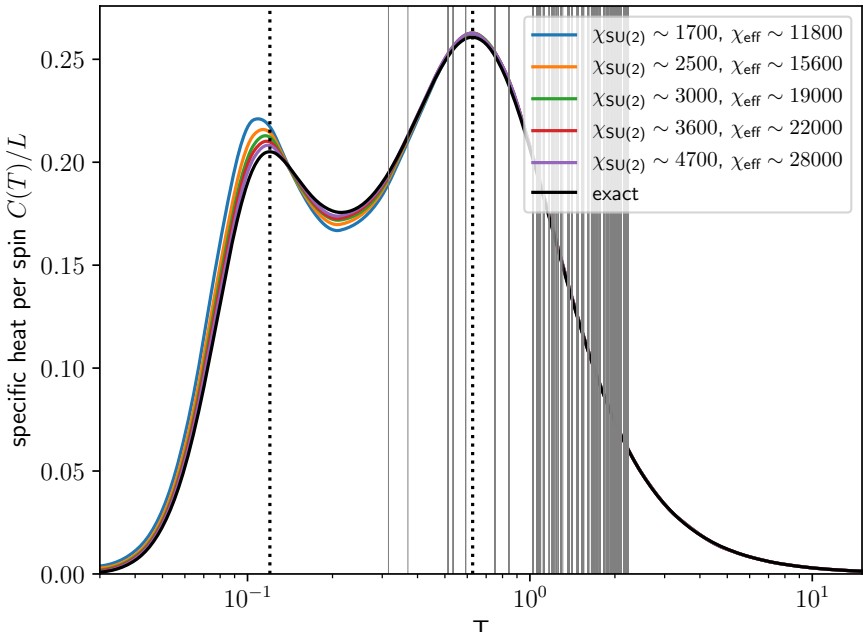

Figure 10: Specific heat of the dodecahedron for different bond dimensions. The bond dimension per subspace was limited to $\chi_{\mathrm{sub,SU(2)}} = 300, 400, 500, 600, 800$. The grey vertical lines indicate the first 1000 eigenenergies relative to the ground state, $E_n - E_0$. The dotted vertical lines indicate the peak positions from Ref. [26]. To obtain the exact result, we used the Kernel Polynomial Method [75] with 1000 lowest eigenstates, 1000 Chebyshev moments and 1000 random vectors.

finite-temperature results are numerically exact, since a much higher bond dimension may be required to achieve such precision, we expect that the qualitative behaviour should be captured as well.

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
