# Peer review of "The antiferromagnetic $S=1/2$ Heisenberg model on the C$_{60}$ fullerene geometry"

_SciPost Physics, doi:SciPost Phys. 10, 087 (2021)_

## Round 1 · Referee Report · Jürgen Schnack (Referee 1) · 2020-11-26

Strengths

I am really impressed by this manuscript. It provides a concise and at the same time rather complete discussion of the magnetic spectrum as well as of the thermodynamic properties of the Heisenberg model on the C60 fullerene geometry. Results are compared with related structures such as the icosahedron or the dodecahedron.

Weaknesses

I suggest to add some recent literature on related investigations of related structures.

Report

In accord with my remarks on strengths and weaknesses I support publication with only minor changes.

Requested changes

References to and discussion of the following literature will further improve this already very good manuscript.

  1. Physics of the related icosidodecahedron (also $I_h$)

  2. a. First DMRG study of such a structure, see M. Exler, J. Schnack, Evaluation of the low-lying energy spectrum of magnetic Keplerate molecules using the density-matrix renormalization group technique, Phys. Rev. B 67 (2003) 094440

  3. b. Please discuss how your results depend on the chosen mapping of C60 onto a chain, i.e. Schlegel graph (line 99) and compare to J. Ummethum, J. Schnack, A.M. Laeuchli, Large-scale numerical investigations of the antiferromagnetic Heisenberg icosidodecahedron, J. Magn. Magn. Mater. 327 (2013) 103-109

1.c. Specific heat of icosidodecahedron, double peak structure(!) J. Schnack, O. Wendland, Properties of highly frustrated magnetic molecules studied by the finite-temperature Lanczos method, Eur. Phys. J. B 78 (2010) 535-541

  1. Please relate your findings shortly to the Cairo pentagonal lattice, see e.g. Quantum magnetism on the Cairo pentagonal lattice, I. Rousochatzakis, A. M. Läuchli, and R. Moessner Phys. Rev. B 85, 104415 (2012)

  2. Line 264: you could relate your results to those obtained recently for a Kagome of size N=42, again double-peak structure of C: J. Schnack, J. Schulenburg, J. Richter, Magnetism of the N=42 kagome lattice antiferromagnet Phys. Rev. B 98 (2018) 094423

  • validity: top
  • significance: high
  • originality: high
  • clarity: top
  • formatting: excellent
  • grammar: excellent

Author:  Roman Rausch  on 2021-02-12  [id 1229]

(in reply to Report 1 by Jürgen Schnack on 2020-11-26)

Dear Prof. Schnack,

Thank you for your very positive review of our work. Our reply is as follows:

>1. a. First DMRG study of such a structure, see
>M. Exler, J. Schnack, Evaluation of the low-lying energy spectrum of magnetic Keplerate molecules using the density-matrix renormalization group technique, Phys. Rev. B 67 (2003) 094440

The icosidodecahedron indeed deserves a comparison, especially since it belongs to the same symmetry group and to the Archimedean solids. This was an oversight on our part. We have now included a comparison in terms of the ground state energy, the gaps and lowest excited states, the correlation functions (in Fig. 5), the specific heat and concluding remarks in Section 5.
Overall, it turns out that its properties are quite different from C60, e.g. it is more strongly frustrated, has a much higher ground state energy, a very small singlet gap, nondegenerate excited states and a 3-peak structure of the specific heat - all in opposition to our case.
On the other hand, the properties of C60 turn out to be quite similar to those of the truncated tetrahedron (C12), which was suspected some time ago. We have therefore also switched out the comparison with the icosahedron for a comparison with C12.

>1. b. Please discuss how your results depend on the chosen mapping of C60 onto a chain, i.e. Schlegel graph (line 99) and compare to
>J. Ummethum, J. Schnack, A.M. Laeuchli, Large-scale numerical investigations of the antiferromagnetic Heisenberg icosidodecahedron,
>J. Magn. Magn. Mater. 327 (2013) 103-109

Since this was also asked by another referee, we have included a detailed discussion in Section 2.1. We also performed a benchmark with the icosidodecahedron and come very close within the exact energy with different numberings. Overall, we come to the same conclusion as the paper you cite: As long as the hopping range is reasonably minimized, convergence is good. An unreasonable numbering (e.g. randomized) is quite detrimental for both MPO compression and convergence; and will probably require a massive amount of bond dimension to compensate. Furthermore, we not that a numbering where the sites i and i+1 are connected helps the TDVP algorithm not to get stuck in the initial state; and we have added a corresponding remark.

>1.c. Specific heat of icosidodecahedron, double peak structure(!)
>J. Schnack, O. Wendland, Properties of highly frustrated magnetic molecules studied by the finite-temperature Lanczos method,
>Eur. Phys. J. B 78 (2010) 535-541

As written above, a discussion has been included. In fact, we see three peaks (also plotted semilogarithmically in: Kunisada & Fukimoto, "Dimer-dimer correlations and magnetothermodynamics of S=1/2 spherical kagome clusters in W72V30 and Mo72V30", 2015).

>2. Please relate your findings shortly to the Cairo pentagonal lattice,
>see e.g.
>Quantum magnetism on the Cairo pentagonal lattice,
>I. Rousochatzakis, A. M. Läuchli, and R. Moessner
>Phys. Rev. B 85, 104415 (2012)

We have included a brief comparison in the introduction. Since the Cairo lattice consists of irregular pentagons with two values of J, its physics seems quite different from C60, whose pentagons are all disjoined.

>3. Line 264: you could relate your results to those obtained recently
>for a Kagome of size N=42, again double-peak structure of C:
>J. Schnack, J. Schulenburg, J. Richter, Magnetism of the N=42 kagome lattice antiferromagnet
>Phys. Rev. B 98 (2018) 094423

This has been included in the revised manuscript.

---

## Round 1 · Referee Report · Anonymous (Referee 2) · 2020-12-21

Strengths

1- State-of-the art numerical results for low-energy and thermodynamic properties of spin-1/2 Heisenberg model on C60 fullerene cluster.

2- Comparison with other frustrated lattices and smaller fullerene geometries

Weaknesses

1- Analysis of numerical data is maybe too detailed at the expense of physical insight.

Report

This article presents a detailed numerical analysis (using DMRG) of the spin-1/2 Heisenberg model on the C60 fullerene geometry. Although DMRG is best suited for one-dimensional geometries, this intermediate-size quantum problem can be solved with sufficient accuracy. The authors have studied both the low-energy spectrum (low-energy excitations, correlations in these low-energy states etc.) as well as thermodynamics (specific heat, susceptibility) that can be compared to other fullerene-like molecules or well-known lattices.

I have few questions and remarks: 1- The physical motivation comes from a Hubbard model on C60 molecule. Since it is in some intermediate-coupling range, the Heisenberg model is the first approximation at half-filling. Is it known if there is a more quantitative effective model to describe the magnetic properties ? 2- When comparing to other lattices, I disagree with the choice of references: (i) for the triangular lattice, it is presumably ordered and proper reference should be made (here only an extended J1-J2 triangular is cited, without any clear reason); (ii) for the kagome lattice, there is already a long list of references but nevertheless, more recent papers pointing towards a gapless Dirac spin liquid should be added. 3- Similarly, in Sec.IV.C, when discussing the triplet gap of the kagome lattice, more caution should be made since it is probably vanishing (gapless Dirac spin liquid). 4- The authors have chosen a 1d path to implement DMRG. How are the results sensitive to it ? Would it be possible to compute at least the energy using a different path ? 5- Section II.C about spin correlations could be shortened (not all data are needed in the text, maybe into some table). Also, the fact that a molecule (with a finite triplet gap) is distinct from a Néel order in an infinite system is quite obvious. There cannot be any symmetry breaking on a finite-size system. 6- Fig.6 is quite puzzling since the triplet state has an inhomogeneous distribution of <Sz> values, while all sites are equivalent. This is clearly an artefact, as stated in the text, pointing to a degenerate manifold. However, the conclusion "judging by (...) other members (...)" is not rigorous enough. Would it be possible to determine in which irrep the lowest triplet transforms ? 7- When computing thermodynamics properties, the authors have chosen to add ancillas, which is one of the standard way to do it. Still, it is non trivial to combine 2- and 1-site TDVP and it would be useful to better explain the choice of parameters and/or provide some benchmarks. 8- Minor points: (i) there are typos (ancilla cites); (ii) Most importantly, the spin gap value of the dedocahedron is incorrect (0.514 rather than 0.519), which I hope is a typo. (iii) the titles in the bibliography should be improved (missing capital letters etc.)

  • validity: high
  • significance: high
  • originality: high
  • clarity: good
  • formatting: excellent
  • grammar: excellent

Author:  Roman Rausch  on 2021-02-12  [id 1230]

(in reply to Report 2 on 2020-12-21)

We would like to thank the referee for the favourable review and the constructive criticism. The raised points are addressed as follows:

>1- The physical motivation comes from a Hubbard model on C60 molecule. Since it is in some intermediate-coupling range, the Heisenberg model is the first approximation at half-filling.
>Is it known if there is a more quantitative effective model to describe the magnetic properties ?

Yes there is: One can take higher terms of the mapping to the Heisenberg model, for example derived by M. Takahashi in "Half-filled Hubbard model at low temperature" (1977). To the order of 1/U^3 one gets a correction to J and a nearest-neighbour term J' proportional to 1/U^3. This is indeed interesting, as it should increase the frustration. The order of 1/U^5 becomes quite difficult to handle, as a biquadratic term is induced.
Now that the properties of the simple Heisenberg model are well-understood, including J' could be done in a future work. We have added some remarks regarding that to Section 5.

>2- When comparing to other lattices, I disagree with the choice of references: (i) for the triangular lattice, it is presumably ordered and proper reference should be made (here only an
>extended J1-J2 triangular is cited, without any clear reason); (ii) for the kagome lattice, there is already a long list of references but nevertheless, more recent papers pointing towards
>a gapless Dirac spin liquid should be added.
>3- Similarly, in Sec.IV.C, when discussing the triplet gap of the kagome lattice, more caution should be made since it is probably vanishing (gapless Dirac spin liquid).

We mentioned the triangular lattice just as a prototypical example of frustration, we did not mean to compare our results to it. The formulation may have been confusing and we have revised it.
Regarding the kagome lattice, a fair comparison should probably be between the finite C60 and a finite kagome cluster that has a finite-size gap. But we agree that the results pointing towards a gapless kagome ground state should also be mentioned, which has been done in the revised manuscript.

>4- The authors have chosen a 1d path to implement DMRG. How are the results sensitive to it ? Would it be possible to compute at least the energy using a different path ?

Since this was also asked by Prof. Schnack, we have included a detailed discussion in Section 2.1. We also performed a benchmark for the smaller icosidodecahedron (30 sites) and come within 99.97% of the ground state energy with different numberings at a bond dimension of 500. Overall, we come to the same conclusion as Ummethum et al. (2013): As long as the hopping range is reasonably minimized, convergence is good. An unreasonable numbering (e.g. randomized) is quite detrimental for both MPO compression and convergence, and will probably require a massive amount of bond dimension to compensate. Furthermore, we note that a numbering where the sites i and i+1 are connected helps the TDVP algorithm not to get stuck in the initial state; and we have added a corresponding remark.
For the C60 case, we calculated the lowest S=1 state using a Cuthill-McKee numbering and find E=-30.7736 (at a bond dimension of 5000), as compared to -30.7756 in the paper (at a bond dimension of 10000).

>5- Section II.C about spin correlations could be shortened (not all data are needed in the text, maybe into some table). Also, the fact that a molecule (with a finite triplet gap)
>is distinct from a Néel order in an infinite system is quite obvious. There cannot be any symmetry breaking on a finite-size system.

We think that understanding the distance behaviour of the correlations is important, as it is needed in order to fit the correlation length, which is a bit tricky for a molecule that has inequivalent bonds at the same distance. But we agree that the discussion turned out to be too long and that repeating the values in the text is not necessary. It has now been shortened to the essential points. We decided not gather the correlation function values into a table, as this would be again a repetition. They can be read off from the plots. On the other hand, we also added a brief comparison to the truncated tetrahedron (C12) and put a list of gaps for various polyhedra in Table 2 for easier reference.
Regarding comparison to Néel order, perhaps our formulation was not clear: The fullerenes have a kind of thermodynamic limit, C_n should approach the hexagonal lattice for n to infinity. While any finite system would not show symmetry breaking, this tendency should become visible from strong spin-spin correlations over long distances. Our observation is merely that C60 is still far away from that limit because such tendency to order is disrupted by the pentagons. We have now improved our formulation.

>6- Fig.6 is quite puzzling since the triplet state has an inhomogeneous distribution of <Sz> values, while all sites are equivalent. This is clearly an artefact, as stated
>in the text, pointing to a degenerate manifold. However, the conclusion "judging by (...) other members (...)" is not rigorous enough. Would it be possible to determine
>in which irrep the lowest triplet transforms ?

We agree that this was a bit too short. We have now included a more thorough discussion and comparison.
Finding out the irrep is possible in principle, if one uses a symmetry-adapted basis within DMRG, but this would require much basic code development and is probably not really worth the trouble. A different way is to construct the whole low-energy multiplet by projecting out each newly found lowest eigenstate. If one can cast a symmetry transformation in an MPO form, one can then diagonalize this operator in the degenerate subspace to find out the right linear combination for a given eigenvalue. Constructing the multiplet is numerically costly and would require the calculation of 4 to 7 eigenstates to an energy precision lower than the next gap. We do not want to attempt such a procedure since there is no pressing physical question associated with the degeneracies of these excited states. Knowing the degeneracy is most important for the ground state, which is clearly nondegenerate and thus belongs to irrep A.
We have added a corresponding discussion to the paper.

>7- When computing thermodynamics properties, the authors have chosen to add ancillas, which is one of the standard way to do it. Still, it is non trivial to combine 2-
>and 1-site TDVP and it would be useful to better explain the choice of parameters and/or provide some benchmarks.

We have now expanded the explanation of our approach in Section 4.1.
The 1- and 2-site TDVP algorithm are in fact easy to combine: At each timestep, one has the choice between the 2-step algorithm, which can increase the bond dimension; or the cheaper 1-step algorithm, which cannot. Since the bond dimension is of course limited from above in the calculations, at some point it makes sense to switch to 1-step for the rest of the propagation.
Another question is if TDVP gets stuck in the product state, as asked by another referee. We have added an explanation of why this does not happen in our case.
In terms of benchmarks, we do provide a benchmark of our approach to limit the bond dimension per subspace rather than the total one in Appendix A, by comparing with C20. If the referee thinks this is insufficient, we would like to know more precisely what would be required. In any case, we have now also added data where the total bond dimension is fixed.

>8- Minor points: (i) there are typos (ancilla cites); (ii) Most importantly, the spin gap value of the dedocahedron is incorrect (0.514 rather than 0.519), which I hope
>is a typo. (iii) the titles in the bibliography should be improved (missing capital letters etc.)

We thank the referee for spotting these typos; they have been corrected.

---

## Round 1 · Referee Report · Anonymous (Referee 3) · 2021-1-21

Strengths

1) the applied DMRG technique is state-of-the-art 2) study of both ground state properties as well as thermodynamics 3) beautiful visualizations

Report

This manuscript studies the properties of the antiferromagnetic Heisenberg model on a C60 fullerene lattice. As electronic properties of the C60 fullerene are an important scientific question, studying the properties of the strong coupling limit at half-filling, described by the antiferromagnetic Heisenberg model is an interesting endeavor. The authors apply the DMRG method employing the full SU(2) symmetry of the model to study ground state, and low-lying excited state properties of the system, and also employ the matrix product state ancilla method to study its thermodynamics. The main results presented are the spin correlation functions of low energy states, specific heat, and magnetic susceptibility.

The technical level of the study is state-of-the-art and the applied SU(2) tensor network technique is impressive. The authors make a great effort to visualize the spin correlation functions, and the more involved thermodynamics simulations appear to be properly carried out. Especially, a proper comparison of the thermodynamic quantities as a function of bond dimension is shown, and convinces that results on the C60 fullerene are converged up to a small residual bias. Also, the bond dimensions reached are noteworthy. While the computational aspect of the manuscript is fully satisfactory, it is not clear what physical conclusions are drawn. Here I would like the authors to address the following questions, before I can recommend the manuscript for publication in SciPost:

(1) What is the nature of the ground state? The manuscript would benefit from a clear answer to this question, if possible. If this is not possible, a discussion of several candidate states is warranted.

(2) How do their results compare to the results in Jiang and Kivelson, Reference [8], on the t-J model? Do the authors' results contribute to developing a clearer picture of the physics in this model? I think it would be beneficial, to add some discussion to compare these two manuscripts.

(3) How can we interpret the particular shape of the specific heat and magnetic susceptibility? In particular, it would be good to discuss the origin of the low energy shoulder of the specific heat. The authors state that the specific heat resembles the kagome lattice specific heat. Is there any reason why these two systems should have similar specific heat?

I also have a few minor remarks:

(a) The authors state in line 175, that the ground state is disordered. However, the lowest-lying excitations is claimed to be a triplet. Would the latter finding not hint towards some magnetic ordering?

(b) It is claimed in line 265, that the kagome antiferromagnet possesses a small neutral gap of size ~0.05, citing two DMRG studies on the subject. To the best of my knowledge, the existence of a singlet gap in this system is still hotly debated, and more recent studies (e.g. He et al., Phys. Rev. X 7, 031020) hint towards a gapless state being realized. Therefore, I think the authors should drop the claim of a singlet gap in the kagome lattice, based on the two cited references.

(c) When performing the thermodynamic simulations, the authors use the two-site TDVP method to perform an imaginary time evolution of an initial infinite temperature state. TDVP suffers from projection error if the initial state has a too small bond dimension. As their results on smaller systems agree nicely with the exact solution, it would be good to mention why the TDVP projection error is negligible.

(d) Since the numerical method applied is quite advanced, it would be interesting whether it could also directly simulate the Hubbard model on the C60 fullerene. Maybe the authors could comment on whether this would currently be possible, or whether further developments are needed for such simulations.

(e) The excited states shown in Fig. 7 and Fig. 6 do not appear to be point group symmetric. This can be expected, since an arbitrary superposition (as obtained from numerics) of the degenerate states need not transform trivially under the point group symmetry. However, this does not imply an additional degeneracy as stated in lines 182 and 183. This raises the question, whether different random initial states for DMRG would yield the same results as shown in Fig. 6 & 7.

(f) The authors state in the abstract that the C20 and C32 molecules are "exactly solvable", and later explain "exactly solvable by numerical diagonalization". I think the term "exactly solvable" should be reserved for analytic solutions. This is of course a very minor remark, and the authors can choose to ignore it.
  • validity: high
  • significance: good
  • originality: good
  • clarity: high
  • formatting: perfect
  • grammar: excellent

Author:  Roman Rausch  on 2021-02-12  [id 1231]

(in reply to Report 3 on 2021-01-21)

We would like to thank the referee for the positive review and the interesting points that have been raised. Our response is as follows:

>(1) What is the nature of the ground state? The manuscript would benefit from a clear answer to this question, if possible. If this is not possible, a discussion of several candidate states is warranted.

We are not aware of such a list of candidate states for molecules. If the list of spin liquid types (e.g. Savary & Balents (2016), Tab. 1) is what the referee has in mind, then they seem not straightforward to extend to finite mesoscopic molecules. These always have a finite-size gap (so that gapless phases disqualify immediately) and their idiosyncratic geometry cannot be simply altered and put on a cylinder, for example.
The best summary we can provide for the ground state is as follows: It is a disordered state with short-range antiferromagnetic correlations, where the weak frustration due to the pentagons prevents quasi-long-range order. It probably sits right at a crossover point between a strongly frustrated molecule (such as the dodecahedron) and a quasi-ordered honeycomb plaquette.

>(2) How do their results compare to the results in Jiang and Kivelson, Reference [8], on the t-J model? Do the authors' results contribute to developing a clearer picture of the physics in this model?
>I think it would be beneficial, to add some discussion to compare these two manuscripts.

Jiang and Kivelson only discuss energy gaps. We notice that their singlet-triplet gap without doping is indeed conistent with ours, as far as we can read it off the plot (about 0.35).
The main point of their paper is the presence of attractive pair binding, but they do not attempt to link it to the geometry. We think this is a point which needs better understanding: Since C60 shows pair binding and the dodecahedron does not [Lin et al., 2007], how can that be related to the geometry? Is the weak frustration an essential part of this? The truncated tetrahedron (C12) also shows attractive pair binding and has similar weakly frustrated properties. We have now included a comparison with it into the revised manuscript instead of the icosahedron.
However, these questions require an understanding of excitations that result from removing electrons rather than flipping spins, i.e. real-space correlations within the t-J model with finite doping, and maybe their dynamics. This is not found in their paper and could be done in a future study. We have included such notes into the outlook.

>(3) How can we interpret the particular shape of the specific heat and magnetic susceptibility? In particular, it would be good to discuss the origin of the low energy shoulder of the specific heat.
>The authors state that the specific heat resembles the kagome lattice specific heat. Is there any reason why these two systems should have similar specific heat?

The specific heat is a function of the distribution of eigenvalues only, and singlets furthermore do not contribute to the susceptibility. A shoulder can only mean a relatively dense clustering of eigenenergies above the gap. We can furthermore say that these eigenenergies should be singlets and triplets due to the fact that the quintet gap is even larger (now added to Tab. 1). But understanding the distribution in more detail is obviously quite difficult unless the full spectrum can be constructed either by exact diagonalization or by an analytical method such as the Bethe ansatz.
Let us further note that the significance of the specific heat is also quantitative: If it can be measured and looks close enough to theory, it would give evidence to the validity of the Heisenberg model description. Furthermore, from the position of the main peak one can estimate J and compare it to J=4t^2/U from other independent estimates of t and U.

The kagome lattice has hexagons surrounded by 6 triangles, whereas C60 has hexagons surrounded by 3 pentagons. This similarity may lead to similar excited states for certain energies (e.g. by flipping spins along the hexagons). Otherwise we cannot think of any geometry on a plane which is a better match for comparison with C60. In terms of the position of the main peak, C60 is also a better match than the icosidodecahedron (now also included into the revised manuscript) which has pentagons surrounded by 3 triangles and was brought up as a molecular analogue of the kagome lattice in the past.
In any case, this is just speculation based on the similar shape of the specific heat and a roughly similar geometry. We do not want to make a very deep point of it, it is just a sidenote. But we believe that the similarity of the specific heat shapes should at least not go unnoticed.

>(a) The authors state in line 175, that the ground state is disordered. However, the lowest-lying excitations is claimed to be a triplet. Would the latter finding not hint towards some magnetic ordering?

As we tried to explain, the ground state is at a crossover point to an ordered (or quasi-ordered) state, with Néel-like correlations along the hexagon bonds. This is what we expect to happen for the larger C70, C80 etc. For C60, the pentagons disrupt the tendency towards order (the exponential dropoff is quite clear, see the inset of Fig. 5 in the new manuscript), yet frustration is also alleviated because they are not joined to each other.

>(b) It is claimed in line 265, that the kagome antiferromagnet possesses a small neutral gap of size ~0.05, citing two DMRG studies on the subject. To the best of my knowledge, the existence of a singlet gap
>in this system is still hotly debated, and more recent studies (e.g. He et al., Phys. Rev. X 7, 031020) hint towards a gapless state being realized. Therefore, I think the authors should drop the claim of
>a singlet gap in the kagome lattice, based on the two cited references.

A fair comparison should probably be between C60 and a finite kagome plaquette which should have a finite-size gap. But we agree that the references in favour of a gapless kagome ground state should be included, which has been done in the revised manuscript.

>(c) When performing the thermodynamic simulations, the authors use the two-site TDVP method to perform an imaginary time evolution of an initial infinite temperature state.
>TDVP suffers from projection error if the initial state has a too small bond dimension. As their results on smaller systems agree nicely with the exact solution,
>it would be good to mention why the TDVP projection error is negligible.

Such a discussion has been included into Section 4.1. We believe that this is no problem in our case, since we use super-sites and our numbering ensures that sites i and i+1 are always connected by a bond.

>(d) Since the numerical method applied is quite advanced, it would be interesting whether it could also directly simulate the Hubbard model on the C60 fullerene.
>Maybe the authors could comment on whether this would currently be possible, or whether further developments are needed for such simulations.

We have included a discussion into the outlook. We did try to solve the full Hubbard model, but find a much larger state error (variance per site), by several orders of magnitude. So it would require even more bond dimension. At the same time, the numerical complexity is larger due to the larger local Hilbert space.

>(e) The excited states shown in Fig. 7 and Fig. 6 do not appear to be point group symmetric. This can be expected, since an arbitrary superposition (as obtained from numerics) of the degenerate states
>need not transform trivially under the point group symmetry. However, this does not imply an additional degeneracy as stated in lines 182 and 183. This raises the question, whether different random
>initial states for DMRG would yield the same results as shown in Fig. 6 & 7.

There seems to be a misunderstanding here: What we meant is exactly what the referee writes: One converges to an arbitrary superposition of the degenerate states. By "additional degeneracy" we meant that this has nothing to do with the threefold triplet degeneracy, which is already fully accounted for in our SU2-symmetric code. We have now added a clarifying remark.
Regarding Fig. 6: In general, random initial states will of course cause a random result. But we find that the 20-site ring is in fact quite robust and is converged to for various initial states and chain numberings. The reasons for that are mysterious to us.

>(f) The authors state in the abstract that the C20 and C32 molecules are "exactly solvable", and later explain "exactly solvable by numerical diagonalization". I think the term "exactly solvable"
>should be reserved for analytic solutions. This is of course a very minor remark, and the authors can choose to ignore it.

We agree and have altered the formulations.

---

## Round 2 · Referee Report · Anonymous (Referee 2) · 2021-2-12

Report

The authors have answered all my comments in a satisfactory manner and have improved their manuscript so I recommend it for publication.

---

## Round 2 · Referee Report · Jürgen Schnack (Referee 1) · 2021-2-12

Report

All suggestions have been incorporated. Please accept manuscript.

---

## Round 2 · Author Response

We would like to thank all the referees for their constructive criticism. We could not do it justice by just changing a sentence here and there, but had to rewrite large parts of the text, update several figures and perform additional calculations.

---

## Round 2 · List of Changes

• improvements of formulations in accordance to the referees' remarks throughout the manuscript
  • corrected typos and notations throughout the manuscript
  • various added references
  • inclusion of Ref. 13 and its mention in Sec. 1
  • comparison with the Cairo tiling (Ref. 22) in Sec. 1
  • mention of the icosidodecahedron and the truncated tetrahedron (C12) in Sec. 1
  • discussion of the mapping to the chain in Sec. 2.1
  • inclusion of C12 and the icosidodecahedron into Sec. 2.2
  • inclusion of the quintet gap into Tab. 1
  • shortening of the discussion of the distance dependence of the correlation functions in Sec. 2.3
  • comparison with the icosidodecahedron and C12 in Sec. 2.3
  • clearer formulation of the significance of the n to infinity limit at the end of Sec. 2.3
  • addition of Tab. 2 (gaps for various polyhedra)
  • the numbering of the sites is now shown in Fig. 4
  • Fig. 5 now contains the icosidodecahedron and C12 instead of the icosahedron
  • Fig. 5 now contains the exponential fit of the distance dependence in the inset
  • more thorough discussion of the irreducible representations in Sec. 3
  • comparison to the icosidodecahedron in Sec. 3
  • more detailed discussion of the TDVP approach in Sec. 4.1
  • the parameters used in the finite-temperature calculations are presented in Tab. 3
  • comparison with the icosidodecahedron and C12 in the discussion of the specific heat (Sec. 4.2)
  • better comparsion to the kagome lattice in Sec. 4.2, mentioning of the possibility of a gapless state, inclusion of the icosidodecahedron
  • the quintet gap and 0.417*triplet gap are shown in Fig. 8
  • icosahedron exchanged for C12 in Fig. 8
  • icosahedron exchanged for C12 in Fig. 9
  • clearer conclusion regarding the questions of frustration vs. order in Sec. 5
  • final comparison with the icosidodecahedron and C12 in Sec. 5
  • mentioning of the Hubbard model and the extended Heisenberg model in Sec. 5
  • comparison to the work by Jiang & Kivelson and resulting implications at the end of Sec. 5

---

## Editorial Decision

published